# Class switching toward IgG4 six months after primary mRNA-based COVID-19 vaccination in kidney patients

**Sophie C. Frölke**[1,2]*, **Kenney G. Amirkhan**[1,3], **Nelly van der Bom-Baylon**[3],
**Marit van Gils**[4], **Mathieu Claireaux**[2,4], **Suzanne E. Geerlings**[2,5,6], **Rory D. de Vries**[7],
**Jan-Stephan F. Sanders**[8], **Luuk B. Hilbrands**[9], **Dimitri A. Diavatopoulos**[10,11],
**A. Lianne Messchendorp**[8,12], **Michiel C. van Aalderen**[1], **Ester B. M. Remmerswaal**[3],
**Frederike J. Bemelman**[1,2]

**1** Amsterdam UMC Location University of Amsterdam, Renal Transplant Unit, Amsterdam, the
Netherlands, **2** Amsterdam Institute for Infection and Immunity, Infectious Diseases, Amsterdam, the
Netherlands, **3** Department of Experimental Immunology, Amsterdam Infection & Immunity, Amsterdam
UMC, University of Amsterdam, Amsterdam, the Netherlands, **4** Department of Medical Microbiology and
Infection Prevention, Laboratory of Experimental Virology, Amsterdam UMC, University of Amsterdam,
Amsterdam, the Netherlands, **5** Amsterdam Public Health, Quality of Care, Amsterdam, the Netherlands,
**6** Amsterdam UMC, Department of Internal Medicine, Location University of Amsterdam, Infectious
Diseases, Amsterdam, the Netherlands, **7** Department of Viroscience, Erasmus University Medical
Center, Rotterdam, the Netherlands, **8** Department of Internal Medicine, Division of Nephrology, University
of Groningen, University Medical Centre Groningen, Groningen, the Netherlands, **9** Department of
Nephrology, Radboud University Medical Centre, Nijmegen, the Netherlands, **10** Radboud Institute for
Molecular Life Sciences, Department of Laboratory Medicine, Laboratory of Medical Immunology, Radboud
University Medical Center Nijmegen, Nijmegen, the Netherlands, **11** Radboud Center for Infectious
Diseases, Radboud University Medical Center Nijmegen, Nijmegen, the Netherlands, **12** Department
of Medical Microbiology and Infection Prevention, University Medical Center Groningen, Groningen, the
Netherlands

* s.c.frolke@amsterdamumc.nl

## Abstract

### Background

Class switching toward spike (S)-binding IgG4 antibodies after mRNA-based COVID-19 vaccination has been observed, an antibody subclass with strong neutralizing but limited effector activity. While this has been reported in healthy individuals, subclass dynamics in immunocompromised kidney patients are unclear. We assessed IgG subclass patterns and S-specific B-cell phenotypes up to 6 months after a two-dose mRNA-1273 vaccination schedule in kidney transplant recipients (KTRs), dialysis patients, and patients with chronic kidney disease (CKD).

### Methods

In this exploratory study, KTRs (n = 11), dialysis patients (n = 5), CKD stage G4–5 patients (eGFR < 30 ml/min/1.73m2, n = 5), and controls without known kidney disease (eGFR > 45 ml/min/1.73m2, n = 8) received two mRNA-1273 doses 28 days apart. Blood was collected pre-vaccination (V1), and at 28 days (V3) and 6 months

org/10.1371/journal.pone.0336320

Clinic Rochester, UNITED STATES OF AMERICA

**Peer Review History:** PLOS recognizes the
benefits of transparency in the peer review
process; therefore, we enable the publication
of all of the content of peer review and
author responses alongside final, published
articles. The editorial history of this article is
available here: https://doi.org/10.1371/journal.
pone.0336320

**Data availability statement:** All relevant data are within the manuscript and its Supporting information files.

**Funding:** This study is funded by The Netherlands Organization for Health Research and Development (ZonMw), project number: 10430072010002. This organization had no role in the study design, data collection and analysis, decision to publish, or preparation of the manuscript.

**Competing interests:** The authors have declared that no competing interests exist.

(V4) after the second dose. S1-specific IgG antibodies were measured by a validated fluorescent bead-based multiplex-immunoassay, and participants seronegative at V1 and seropositive at V3 were included. B cells were phenotyped by flow cytometry.

## Results

Five of 11 KTRs had no detectable S-binding B cells, whereas all other groups mounted responses. Across responders, the frequency of S-binding B cells increased from V1 (median 0.08%) to 0.49% at V3 and to 0.84% at V4 (both $p < 0.0001$). S-binding B cells mainly comprised $IgG^+$ plasmablasts. The IgG4:IgG1 log-ratio increased significantly from V3 to V4 ($p < 0.001$), indicating a relative shift toward IgG4; absolute frequencies were comparable across the groups.

## Conclusions

Approximately half of KTRs lacked detectable S-binding B cells after two mRNA-1273 doses, despite antibody formation. Among responders, S-binding B cells persisted up to 6 months after vaccination with a relative shift toward IgG4, a pattern also observed in dialysis patients, CKD patients and controls. The clinical significance of this subclass skewing requires confirmation in larger cohorts with functional antibody readouts.

## Introduction

End-stage renal disease (ESRD) is associated with accelerated immune senescence and chronic inflammation, rendering patients immunocompromised [1]. COVID-19 vaccination induces protective antibodies in this population, but immunogenicity is reduced compared to controls [2], necessitating booster doses to counter declining antibody levels and emerging viral variants. Kidney transplant recipients (KTRs) are further immunocompromised due to lifelong immunosuppressive therapy and mount weaker vaccine responses than healthy individuals, with up to 50% failing to seroconvert after primary vaccination [2–5]. Repeated vaccination has therefore been recommended for this group [6].

IgG4 is the least abundant IgG subclass in humans and is characterized by unique properties: limited capacity to engage Fc-dependent effector pathways but the ability to generate high-affinity neutralizing antibodies [7–9]. In the context of COVID-19 vaccination, higher concentrations of neutralizing antibodies correlate with reduced risk of infection and disease [2,10–13]. Repeated mRNA-based COVID-19 vaccination has been associated with changes in antibody subclass distribution. In healthy individuals, a third dose can lead to a delayed increase of spike (S)-specific IgG4 antibodies. Similar findings were observed in ESRD patients, where a high fraction of S-binding IgG consisted of IgG4, paralleled by increases in serum IgG4, emerging after the third and fourth doses [9]. In KTRs, IgG4 induction was first observed two months after the fourth vaccination, whereas measurements after the second

vaccination were limited to an early time point (21 days) [14]. This suggests that IgG4 induction requires repeated or sustained antigen exposure [15–17]. Whether ESRD patients and KTRs undergo IgG4 class switching at later time points after the primary two-dose regimen remains unknown.

Here, we compared S-specific humoral responses in KTRs and ESRD patients versus non-immunocompromised controls six months after two mRNA-1273 doses. Specifically, we investigated whether IgG4 class switching occurs under immunosuppressive therapy, to provide insight into vaccine-induced immunity in immunocompromised populations.

## Materials and methods

This study was performed in participants of the RECOVAC Immune Response (IR) study, conducted between February 1 and May 31, 2021, who visited the outpatient clinic of Amsterdam UMC in the Netherlands. The design and results of the RECOVAC-IR study have been published previously [2,18]. All participants were aged ≥ 18 years and provided written informed consent. The study was approved by the Dutch Central Committee on Research Involving Human Subjects (CCMO, NL76215.042.21) and the local Ethics Committee of Amsterdam UMC, and was conducted in accordance with the Declaration of Helsinki. The trial was registered at ClinicalTrials.gov (NCT04741386) and funded by The Netherlands Organization for Health Research and Development (ZonMW, project number 10430072010002).

### Study group

Four groups were included in this study: (i) kidney transplant recipients (KTRs), (ii) patients on hemodialysis or peritoneal dialysis (HD/PD), (iii) patients with advanced chronic kidney disease (CKD; eGFR < 30 mL/min/1.73m², stages G4/5), and (iv) non-immunocompromised controls (CTRLs) without kidney disease (eGFR > 45 mL/min/1.73m²). The control group consisted of partners, siblings, or household members of participants from the patient groups. All participants received two doses of mRNA-1273 (Moderna Biotech Spain, S.L.), administered 28 days apart in accordance with the manufacturer's instructions. Blood samples were collected at baseline (before the first vaccination, V1), 28 days after the second vaccination (V3), and 6 months after the second vaccination (V4). Baseline S1-specific IgG antibodies were measured to exclude individuals with prior SARS-CoV-2 infection. During the study, SARS-CoV-2 infection was defined as either a self-reported positive PCR test or the presence of nucleocapsid-specific antibodies. Being classified as a responder, defined by seroconversion at V3 with a threshold for seropositivity of S1-specific IgG antibody concentrations ≥10 BAU/mL based on receiver operator curve analysis [2], was an inclusion criterion, ensuring the highest likelihood of detecting S-binding B cells. Participants were selected from the four RECOVAC groups with a distribution proportional to each group size.

As negative controls, six individuals without an antibody response were analyzed; all of these were KTRs. These antibody non-responders had ≤ 15 S-binding B cells per ~2 × 10⁶ total cells at both V3 and V4. Therefore, of the 29 eligible participants who were initially selected, participants with ≤15 S-binding B cells at V3 and V4 and/or lower counts at V3 than at V1 were excluded. This resulted in the removal of five individuals, all of whom were KTRs. Finally, six KTRs (KTR1–6) were included, leaving 24 participants for analysis (Table 1). The median age was 56 years, and 58% were female.

### Isolation of PBMCs, flow cytometry strategy and antibody measurements

PBMCs were isolated from peripheral blood by density gradient centrifugation and cryopreserved until analysis as described previously [19,20]. The S-protein was manufactured as described elsewhere [21,22]. B-cell tetramers (S-baits) were generated using Streptavidin AF647 and Streptavidin BV421, incubated with S-protein in four 10-min rounds (total 60 min) at 4°C in the dark, followed by addition of biotin. Thawed PBMCs were washed in PBS/0.01% NaN₃/0.5% BSA /2mM EDTA (FACS buffer) and incubated with S-baits for 30 min at 4°C in the dark. Subsequently, cells were stained with surface monoclonal antibodies (mAbs) in FACS buffer at concentrations according to the manufacturer's instructions and incubated in the dark for 30 minutes at 4°C (S1 Table). Fixable Viability Dye eFluor™ 506 (Invitrogen), CD14 and CD3 were included to exclude dead cells and non-B cells. We detected a fraction of IgG B cells that was not IgG1, IgG2

**Table 1. Baseline characteristics per study group.**

| | CTRL (n=8) | CKD (n=5) | HD/PD (n=5) | KTR (n=6) |
|---|---|---|---|---|
| Female, n (%) | 7 (87.5) | 1 (20.0) | 4 (80.0) | 2 (33.3) |
| Age (years) | 57.2 (12.1) | 55.8 (16.2) | 52.8 (11.5) | 55.2 (11.5) |
| BMI (kg/m²) | 30.0 (7.3) | 28.2 (4.8) | 22.4 (3.5) | 25.0 (3.5) |
| Number of comorbidities | 0 (0, 1) | 2 (2, 3) | 1 (1, 1) | 1 (1, 2) |
| Comorbidities, n (%) | | | | |
| Hypertension | 0 (0.0) | 4 (80.0) | 4 (80.0) | 6 (100.0) |
| Diabetes Mellitus | 2 (25.0) | 3 (60.0) | 0 (0.0) | 0 (0.0) |
| History of coronary artery disease | 0 (0.0) | 3 (60.0) | 1 (20.0) | 0 (0.0) |
| Heart failure | 0 (0.0) | 1 (20.0) | 1 (20.0) | 0 (0.0) |
| Chronic lung disease | 1 (12.5) | 0 (0.0) | 0 (0.0) | 1 (16.7) |
| History of malignancy[1] | 1 (12.5) | 0 (0.0) | 0 (0.0) | 1 (16.7) |
| Primary renal diagnosis, n (%) | | | | |
| Primary glomerulonephritis | - | 0 (0.0) | 1 (20.0) | 1 (16.7) |
| Congenital diseases | - | 1 (20.0) | 1 (20.0) | 2 (33.3) |
| Vascular diseases | - | 2 (40.0) | 2 (40.0) | 0 (0.0) |
| Secondary glomerular/systemic disease | - | 1 (20.0) | 0 (0.0) | 1 (16.7) |
| Other | - | 1 (20.0) | 0 (0.0) | 2 (33.3) |
| Unknown | - | 0 (0.0) | 1 (20.0) | 0 (0.0) |
| Dialysis characteristics | | | | |
| Hemodialysis, n (%) | - | - | 3 (60.0) | - |
| Peritoneal dialysis, n (%) | - | - | 2 (40.0) | - |
| Time on dialysis (years) | - | - | 2.5 [1.9, 5.4] | - |
| Transplant characteristics | | | | |
| First kidney transplant, n (%) | - | - | - | 5 (83.3) |
| Time after last transplantation (years) | - | - | - | 18.7 [10.4, 26.1] |
| Living, n (%) | - | - | - | 4 (66.7) |
| Pre-emptive, n (%) | | | | 2 (33.3) |
| Number of immunosuppressive agents | - | - | - | 3 (2, 3) |
| Immunosuppressive treatment, n (%) | | | | |
| Steroids | - | - | - | 6 (100.0) |
| Azathioprine | - | - | - | 1 (16.7) |
| Mycophenolate mofetil | - | - | - | 2 (33.3) |
| Calcineurin inhibitor | - | - | - | 6 (100.0) |
| eGFR (mL/min/1.73m²) | 84.9 (18.5) | 19.2 (6.1) | - | 42.4 (13.4) |
| Lymphocytes (10⁹/L) | 0.8 (0.3) | 0.5 (0.2) | 0.1 (0.4) | 0.1 (0.5) |

Variables are presented as mean ± SD, or as median (IQR) in case of non-normal distribution.

[1]Including melanomas, excluding all other skin malignancies.

*Abbreviations:* CTRL, control; CKD, chronic kidney disease; HD/PD, dialysis; KTR, kidney transplant recipient; BMI, body mass index; eGFR, estimated glomerular filtration rate.

or IgG3 positive, and this fraction was interpreted as being IgG4. The gating method of IgG4 was therefore exclusionary. This fraction within the S-binding IgG B cell population, identified using our B cell phenotyping flow cytometry panel, has previously been shown to correspond with serum IgG4 levels, thereby justifying its use for the detection of S-binding IgG4 B cells [9]. After surface staining, fixation and permeabilization (eBioscience) were performed prior to intracellular staining

with Ki67 for 30 minutes at 4°C, in the dark. Samples were acquired on a LSR-Fortessa (BD Biosciences) and analyzed with FlowJo v10.10.0. Gating strategy is shown in S1 Fig. Frequencies of S-binding B cells were always presented alongside the entire phenotype-matched switched memory B cell population. Lastly, S1-specific IgG antibodies were measured in serum samples by a validated fluorescent bead-based multiplex-immunoassay with a specificity and sensitivity of 99.7% and 91.6%, respectively [2].

## Statistical analysis

As this was an exploratory study, no formal sample size calculation was performed. Variables are presented as mean ± standard deviation (SD) if normally distributed, or as median with interquartile range (IQR) if non-normally distributed. For comparisons between two independent groups, the Mann–Whitney U test was used, with effect sizes calculated as $r = Z/\sqrt{N}$, where Z is the standardized test statistic and N is the total sample size. Differences between multiple independent groups were assessed with the Kruskal–Wallis test, with effect size expressed as epsilon-squared $(\varepsilon^2) = H/(N-1)$, where H is the test statistic and N the total sample size. Paired comparisons between two related samples were conducted using the Wilcoxon matched-pairs signed-rank test, with effect sizes calculated as $r = Z/\sqrt{n}$, where Z is the standardized test statistic and n is the number of paired observations. For comparisons across multiple related time points within groups, the Friedman test was applied, with effect size expressed as Kendall's $W = \chi^2/[n \cdot (k-1)]$, where $\chi^2$ is the Friedman statistic, n the number of subjects, and k the number of time points. Correlations were analyzed using Spearman's rank correlation coefficient. Statistical significance is denoted as *p < 0.05, **p < 0.01, ***p < 0.001, and ****p < 0.0001. Analyses and graphing were performed with GraphPad Prism version 9.5.1. Detailed statistical results are provided in the figure legends and supplemental material.

## Results

### S-binding B cells are detectable in KTRs, HD/PD, and CKD patients

First, we analyzed S1-specific IgG antibody responses. All participants were seronegative at baseline and seropositive after vaccination, as per the inclusion criteria. As expected, overall antibody levels declined between 28 days and 6 months post-vaccination: from a median of 1575 BAU/mL (IQR 616.5–3386) at V3 to 237.9 BAU/mL (IQR 105.8–492.7) at V4 (p < 0.001, r = −0.78) (Fig 1a) [2,23]. At V3, antibody levels appeared lower in KTRs compared to all other groups, reaching statistical significance only in comparison with CKD patients, while no significant differences were observed at V4. All groups showed a decline in antibody levels from V3 to V4, which was statistically significant in CTRLs and showed a trend in CKD and HD/PD patients, but not in KTRs. In parallel, we investigated whether S-binding B cells were detectable in KTRs, HD/PD and CKD patients, and controls. As a starting point, we measured the frequency of switched memory B cells (defined as IgD⁻CD27⁺) among total CD19⁺B cells (S2 Fig). The percentage of switched memory B cells did not differ significantly between V1, V3, and V4, either overall, or within individual study groups. Next, we assessed the percentage of S-binding B cells within the switched memory B cell population. Five participants lacked detectable S-binding B cells, despite antibody formation; all of them were KTRs. All other groups mounted responses. These five KTRs did not differ from the remaining six KTRs with detectable S-binding B cells in terms of immunosuppressive regimen (S2 Table). At V3, S-binding non-responders showed a lower percentage of lymphocytes (median 70.8; IQR 68.1–71.5) compared with responders (median 76.5; IQR 72.7–81.4; p = 0.015, r = −0.44), although this difference did not persist at V4 (p = 0.276, r = −0.21). In addition, non-responders had a lower percentage of naive B cells at V3 (median 34.6; IQR 15.5–41.9) compared with responders (median 59.2; IQR 41.0–74.0; p = 0.009, r = −0.47), and this difference persisted at V4 (p = 0.023, r = −0.42). These five participants were excluded from subsequent analyses; results below therefore reflect only S-binding responders (Fig 1b). Overall, the frequency of S-binding B cells increased significantly over time when compared to baseline (V1, median 0.08; IQR 0.04–0.14; V3 (28 days), median 0.49; IQR 0.32–0.81; p < 0.0001, r = 0.63; V4 (6 months), median 0.84; IQR 0.42–1.27; p < 0.0001, r = 0.81). S-binding B cell frequencies did not increase further

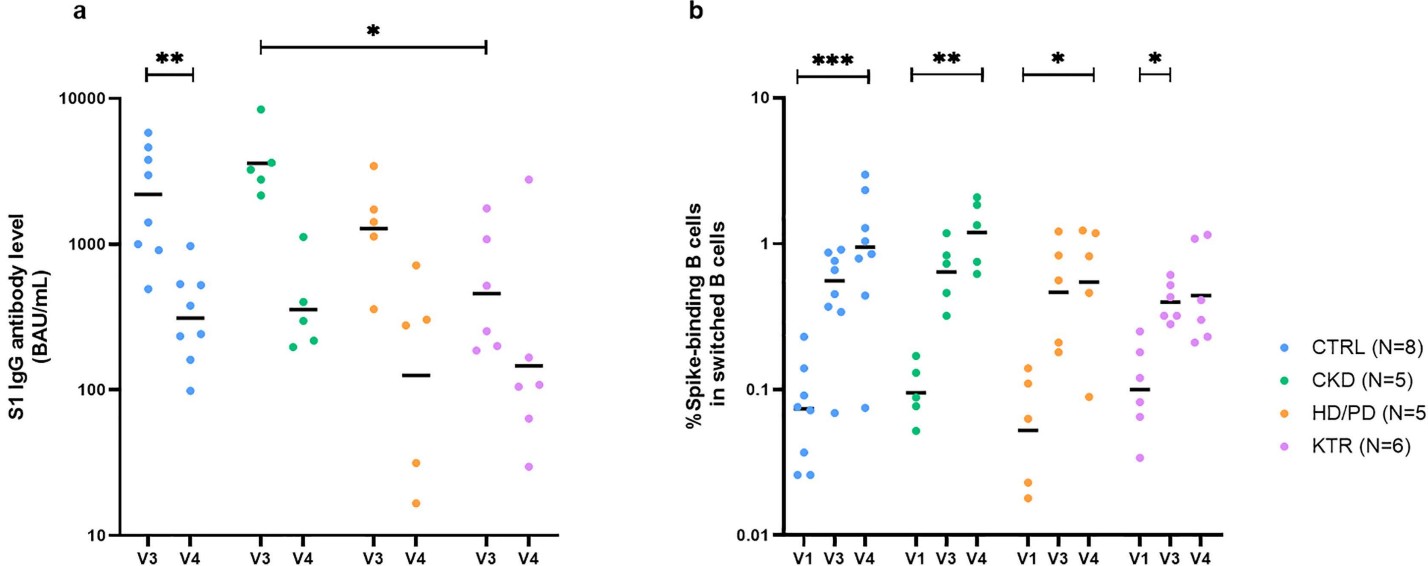

**Fig 1. S1 IgG antibody levels and S-binding B cell frequencies over time.** (**a**) S1 IgG antibody levels in CTRL, CKD, HD/PD, and KTR groups at 28 days post-vaccination (V3), and 6 months post-vaccination (V4). Data are shown on a $\log_{10}$ scale; horizontal lines indicate geometric means. At V3, levels in KTRs were significantly lower than in CKD patients (p = 0.017, r = 0.90), but not different from those in CTRLs (p = 0.165, r = 0.59) or HD/PD patients (p > 0.999, r = 0.41). At V4, no significant between-group differences were observed (p = 0.325, ε² = 0.15). (**b**) Frequencies of S-binding B cells in the same groups at baseline (V1), V3, and V4. Data are shown on a $\log_{10}$ scale; horizontal lines indicate geometric means. No significant between-group differences were observed at any time point (V1 p = 0.509, ε² = 0.10; V3 p = 0.482, ε² = 0.11; V4 p = 0.235, ε² = 0.19).

during follow-up (p = 0.582, r = 0.19). No significant differences between groups were observed at any time point. In the KTR group, the percentage of S-binding B cells was significantly higher at V3 but not at V4 when compared to V1. Within the HD/PD, CKD, and CTRL group, this increase was evident at V3 as well as V4 when compared to V1. The S-binding B cell count at V1 was, in line with the inclusion criteria, considered negligible for all participants.

Medians, IQRs, and full statistical comparisons are provided in S3 Table.

Taken together, almost half of KTRs lacked detectable S-binding B cells despite antibody formation, whereas all other groups mounted responses. Across S-binding responders, antibody titers declined during the follow-up period, and S-binding B cell frequencies were elevated relative to baseline in a similar pattern across groups.

## S-binding B cells display a proliferative plasmablast phenotype in KTRs, HD/PD, and CKD patients

Next, we assessed whether S-binding B cells exhibited features of antibody-secreting cells (ASCs, defined as CD38⁺⁺CD24⁻) in KTRs, HD/PD and CKD patients, and controls. ASCs were identified within both the S-binding and entire switched memory B-cell populations (Fig 2a). No significant differences were observed between groups at V3 or V4, nor within groups over time. In CTRLs, ASC frequencies within the S-binding population were higher than within the overall switched memory pool at both time points. To further characterize ASCs, we examined their composition of plasmablasts (Ki67⁺) versus plasma cells (Ki67⁻) [24]. The majority of S-binding ASCs were plasmablasts at both V3 (median 76.9%, IQR 66.7–92.2) and V4 (67.7%, IQR 50.0–84.8) (p = 0.1261, r = −0.33) (S3 Fig). This pattern was also observed within the switched population with a median proportion of 85.90% (IQR 77.25–88.90) at V3 and 83.35% (IQR 78.90–88.13) at V4 (p = 0.555, r = −0.09).

In addition, CXCR3 expression, consistent with responsiveness to CXCL9 and CXCL10 [25], was assessed within both the S-binding and entire switched memory B cell populations (Fig 2b). CXCR3 expression was detectable at both V3 and

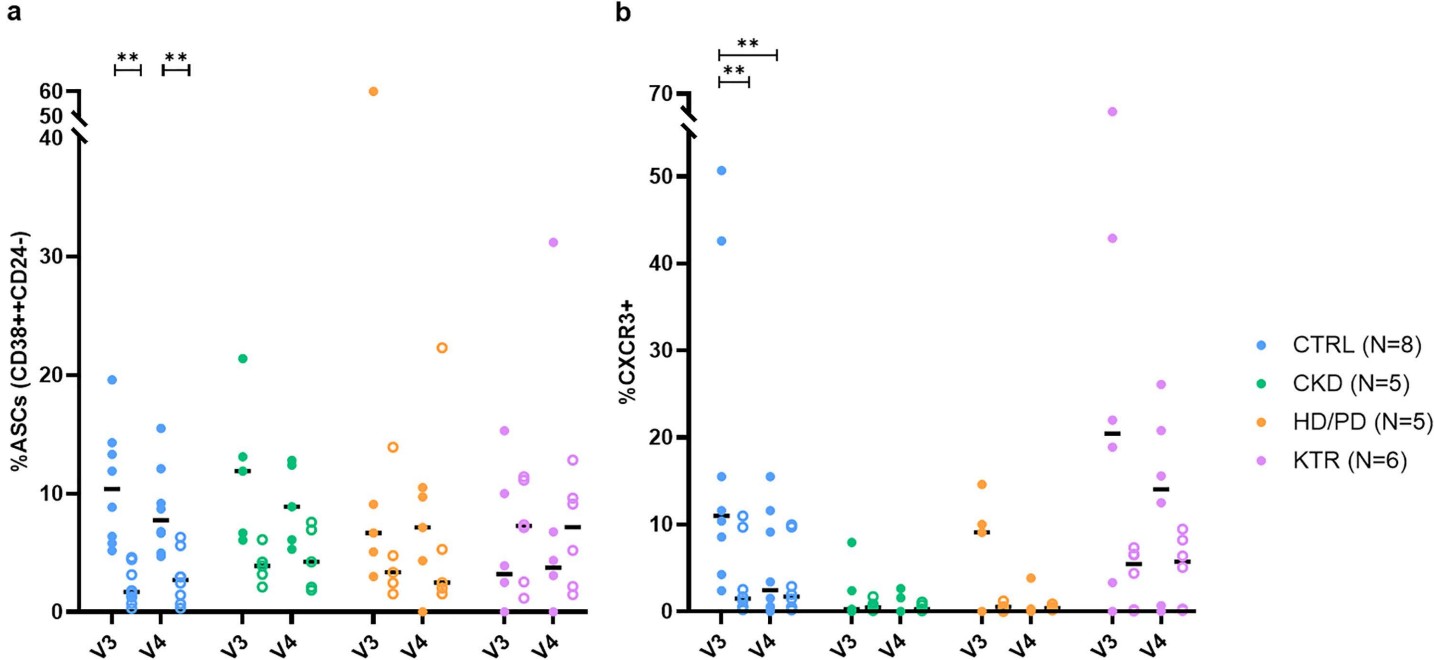

**Fig 2. Circulating ASCs and CXCR3⁺B cells within S-binding and switched memory B cell populations. (a)** Frequencies of antibody-secreting cells (ASCs) within S-binding B cells (filled circles) and within switched memory B cells (open circles) at 28 days (V3) and 6 months (V4) post-vaccination in CTRL, CKD, HD/PD, and KTR groups. Horizontal lines indicate average percentages. Between-group comparisons showed no significant differences at V3 (p=0.239, $\varepsilon^2$=0.18) or V4 (p=0.314, $\varepsilon^2$=0.15). **(b)** Frequencies of CXCR3⁺cells within S-binding B cells (filled circles) and within switched memory B cells (open circles) at V3 and V4 in the same groups. Horizontal lines indicate average percentages. Between-group differences were not significant at V3 (p=0.050, $\varepsilon^2$=0.34) or V4 (p=0.058, $\varepsilon^2$=0.33).

V4 across all groups and in both cell populations. No statistically significant between-group differences were observed; only in CTRLs were S-binding CXCR3 frequencies higher at V3 than at V4. Within-group comparisons showed higher percentages of S-binding cells in CTRLs compared to switched cells at V3, but not at V4.

Medians, IQRs, and full statistical comparisons are provided in S4 Table.

In summary, among S-binding responders, S-binding B cells with an ASC phenotype were detectable across groups and were predominantly plasmablast-like. CXCR3 expression was detectable at both time points, without clear differences between the groups.

## S-binding B cells predominantly express IgG in KTRs, HD/PD, and CKD patients

Next, we examined Ig isotype expression on S-binding B cells in KTRs, HD/PD and CKD patients, and controls. All three isotypes—IgM, IgA, and IgG—were detectable across groups and time points (Fig 3). Overall, at V3, the median percentages (IQR) of isotype expression within S-binding B cells were 7.52% (3.58–21.22) for IgM, 9.88% (6.79–16.05) for IgA, and 68.95% (58.65–78.03) for IgG. The proportion of S-binding B cells expressing IgG was significantly higher compared to both IgM (p<0.0001, r=0.82) and IgA (p<0.0001, r=0.76), while the proportion of S-binding B cells expressing IgM or IgA did not differ (p>0.999, r=0.06). At V4, median percentages (IQR) were 7.22% (4.24–16.17) for IgM (comparable to V3, p=0.966, r=−0.01), 4.36% (2.64–7.71) for IgA (significantly lower than at V3, p=0.001, r=−0.62), and 70.40% (60.65–78.30) for IgG (comparable to V3, p=0.261, r=0.23). Again, proportions of S-binding B cells expressing IgG were significantly higher compared to both IgM (p<0.0001, r=0.71) and IgA (p<0.0001, r=0.94), while no significant difference was observed between IgM and IgA (p=0.372, r=0.22). Between and within groups, the proportion of IgM⁺S-binding

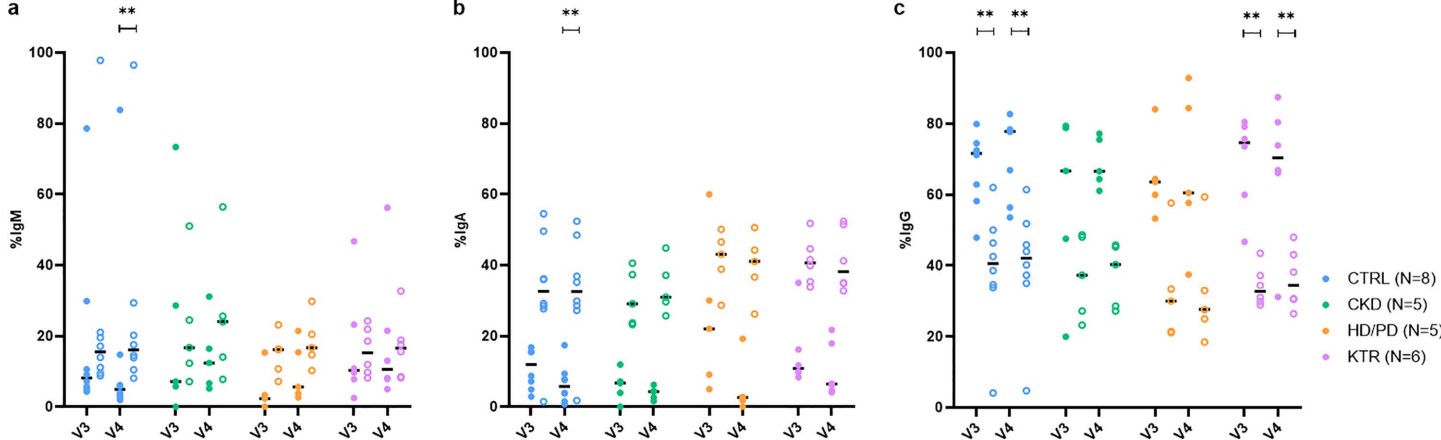

**Fig 3. Frequencies of IgM⁺, IgA⁺, and IgG⁺B cells within S-binding and switched memory B cell populations. (a)** Frequencies of IgM⁺B cells within S-binding B cells (filled circles) and within switched memory B cells (open circles) at 28 days (V3) and 6 months (V4) post-vaccination in CTRL, CKD, HD/PD, and KTR groups. Horizontal lines indicate average percentages. Between-group comparisons showed no significant differences at V3 ($p = 0.177$, $\varepsilon^2 = 0.10$) or V4 ($p = 0.300$, $\varepsilon^2 = 0.03$). **(b)** Frequencies of IgA⁺B cells within S-binding and switched memory B cell populations at V3 and V4 in the same groups. Horizontal lines indicate average percentages. Between-group differences were not significant at V3 ($p = 0.102$, $\varepsilon^2 = 0.16$) or V4 ($p = 0.182$, $\varepsilon^2 = 0.09$). **(c)** Frequencies of IgG⁺B cells within S-binding and switched memory B cell populations at V3 and V4 in the same groups. Horizontal lines indicate average percentages. Between-group differences were not significant at V3 ($p = 0.871$, $\varepsilon^2 = 0.00$) or V4 ($p = 0.900$, $\varepsilon^2 = 0.00$).

B cells, typically associated with the early immune response, was comparable (Fig 3a). Following this early phase, class-switching to IgA or IgG may occur. The proportions of IgA⁺ (Fig 3b) and IgG⁺ (Fig 3c) S-binding B cells did not differ significantly between or within groups. In CTRLs, frequencies of IgM⁺ and IgA⁺S-binding B cells were lower than IgM⁺ and IgA⁺ switched cells at V4, whereas frequencies of IgG⁺S-binding cells were higher than IgG⁺ switched cells at both V3 and V4. In KTRs, frequencies of IgG⁺S-binding cells also exceeded frequencies of IgG⁺ switched cells at V3 and V4.

Medians, IQRs, and full statistical comparisons are provided in S5 Table.

In conclusion, among S-binding responders, S-binding B cells expressed IgM, IgA, and IgG isotypes in comparable patterns across KTRs, HD/PD patients, CKD patients, and CTRLs, with IgG being the dominant isotype at both early and late post-vaccination time points.

## S-binding IgG4 is detectable in KTRs, HD/PD, and CKD patients and increases over time

Next, we examined the IgA and IgG subclass distribution within S-binding B cells (Fig 4a). IgA1 and IgG1 predominated at both time points. IgA1 and IgA2 frequencies remained stable, whereas IgG subclass analysis revealed a marked temporal switch. From V3 to V4, IgG1⁺S-binding B cell frequencies remained stable, IgG2⁺ and IgG3⁺declined, and IgG4⁺increased significantly. This enrichment of IgG4 was inversely associated with IgG1 frequencies (Spearman $r = -0.77$, $p < 0.0001$), but not with IgG2 (Spearman $r$: $-0.142$, $p = 0.510$) or IgG3 (Spearman $r$: $-0.171$, $p = 0.426$) (Fig 4b). A log-ratio analysis confirmed a significant increase in IgG4:IgG1 from V3 to V4 ($p = 0.0002$, $r = 0.77$), indicating a relative shift from IgG1 toward IgG4. This increase in IgG4 was not observed in the memory switched B cell population, which was relatively stable between V3 with a median percentage (IQR) of 5.79% (4.20–11.38) and V4 with 5.94% (4.07–10.58) ($p = 0.900$, $r = 0.03$) (IgG4:IgG1 log-ratio from V3 to V4, $p = 0.790$, $r = -0.06$).

Medians, IQRs, and full statistical comparisons are provided in S6 Table.

No difference existed in S-binding IgA and IgG subclasses between groups at V3 or V4 (Fig 5). Within-group comparisons revealed only in CTRLs a decline of IgG2 and IgG3 over time, as IgG4 proportions increased. When performing the same log-ratio analysis within subgroups, only in CTRLs a significant increase in IgG4:IgG1 was observed ($p = 0.0234$,

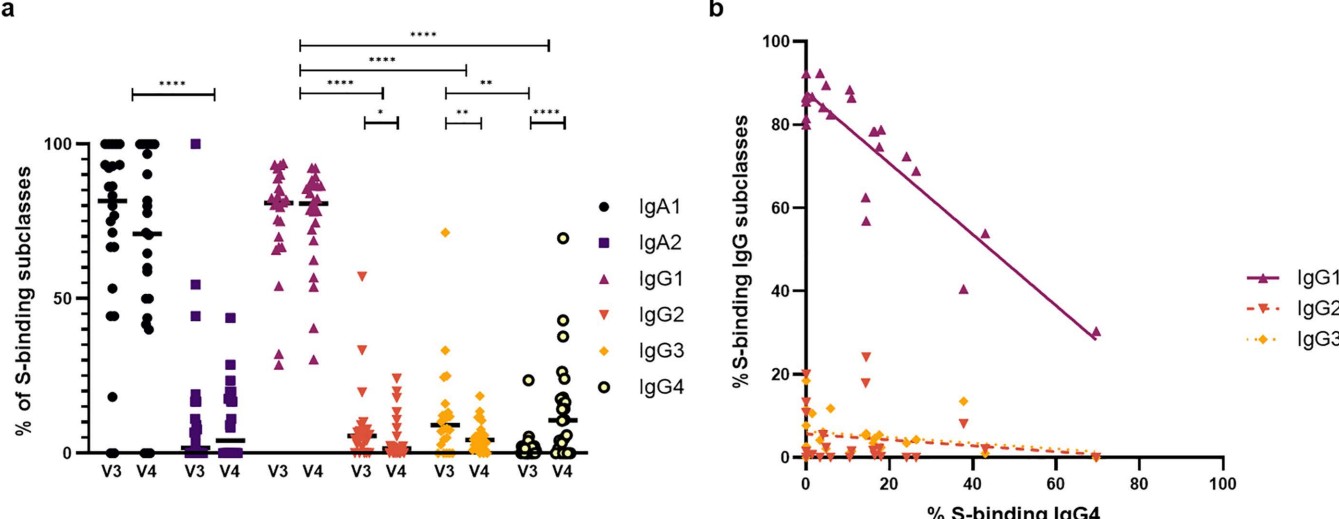

**Fig 4. Frequencies of IgA⁺ and IgG⁺ S-binding B cell subclasses (total 100%) and correlations of IgG4 with other IgG subclasses. (a)** Frequencies of IgA1⁺ and IgA2⁺ S-binding B cells and IgG1–4 subclass distributions at V3 (28 days) and V4 (6 months) post-vaccination in CTRL, CKD, HD/PD, and KTR groups. Horizontal lines indicate average percentages. Between-subclass comparisons showed IgA1>IgA2 at both time points and IgG1 consistently higher than other subclasses. Longitudinal analysis showed stable IgG1 expression over time (p=0.861, r=−0.04), a significant decrease in IgG2 (p=0.022, r=−0.47) and IgG3 (p=0.001, r=−0.66), and a marked increase in IgG4 (p<0.0001, r=0.80). **(b)** Correlation analysis at V4 showed an inverse association between IgG4 and IgG1 (Spearman R: −0.7710, p<0.0001), but no association with IgG2 (Spearman R: −0.142, p=0.510) or IgG3 (Spearman R: −0.171, p=0.426).

r=0.79). No significant changes were detected in CKD (p=0.1250, r=0.92) or HD/PD patients (p=0.500, r=0.95), or KTRs (p=0.1875, r=0.66). Among these KTRs, four of six developed IgG4 responses despite immunosuppressive therapy with corticosteroids and calcineurin inhibitors, with two also receiving MMF (S2 Table, S4 Fig). The two without IgG4 expression received similar regimens, including one on azathioprine.

Frequencies of IgA1⁺, IgA2⁺ and IgG1–4 B cell subclasses at V3 (28 days) and V4 (6 months) post-vaccination in CTRL, CKD, HD/PD, and KTR groups. No significant between-group differences were observed.

Medians, IQRs, and full statistical comparisons are provided in S7 Table.

In summary, among S-binding responders, S-binding B cells predominantly expressed IgA1 and IgG1, but a delayed shift toward IgG4 was observed across groups. Importantly, IgG4 induction was detectable in KTRs, despite immunosuppressive therapy.

## Discussion

In this study, we provide exploratory evidence suggesting that IgG4 class switching can occur after a two-dose mRNA-1273 regimen, including in KTRs under immunosuppression. Nearly half of KTRs lacked detectable S-binding B cells after vaccination, despite antibody formation. However, S-binding responders showed durable B-cell responses, including memory B-cell expansion and plasmablast formation, comparable to those in dialysis patients, CKD patients, and controls.

S-binding non-responders showed lower lymphocyte percentages, consistent with B cell lymphopenia previously described in transplant recipients [26,27]. They also had reduced frequencies of naive B cells, whereas a larger naive B cell pool has been associated with successful humoral responses [26]. As novel immune responses originate from the naive pool, these findings suggest that an insufficient precursor repertoire may limit vaccine-induced immunity in KTRs. Among KTRs with detectable S-binding B cells, frequencies were comparable between groups at both 28 days and 6

## S-binding B cell subclasses

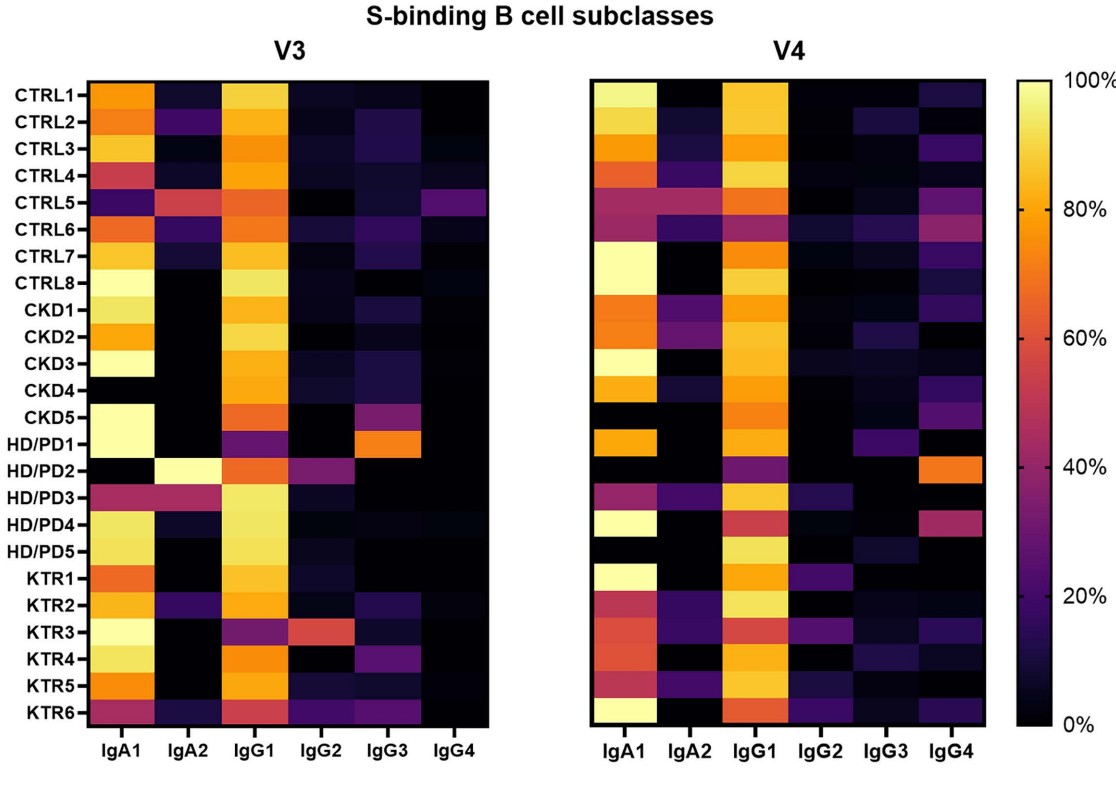

## B cell subclasses in switched cells

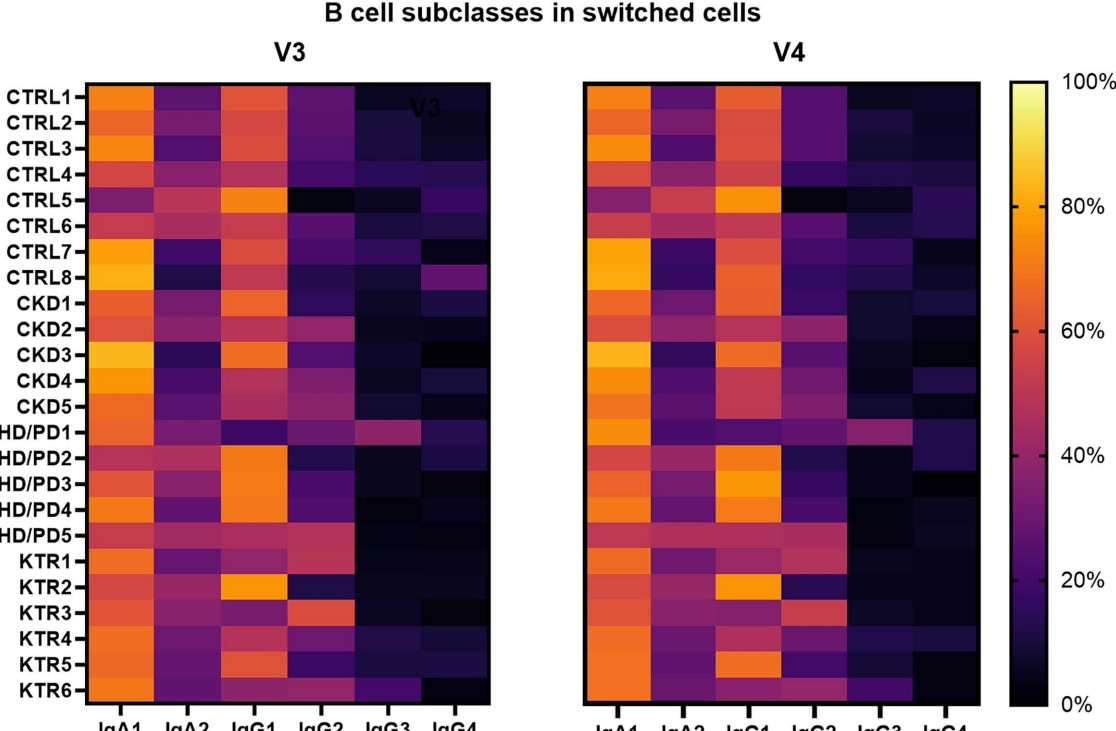

**Fig 5. Heatmaps showing IgA and IgG subclasses (total 100%) within S-binding and switched memory B cell populations.**

months post-vaccination. In line with observations in healthy individuals [28,29], S-binding B cells remained detectable up to 6 months and increased over time despite declining antibody titers, likely reflecting the accumulation of memory B cells [30]. The phenotypic similarity of S-binding B cells in KTRs to those in other groups suggests that their presence could serve as a reliable surrogate marker of immunity. Notably, these cells predominantly exhibited an antibody-secreting plasmablast phenotype, whereas only a smaller fraction were plasma cells, consistent with the notion that long-lived plasma cells predominantly reside in the bone marrow.

S-binding B cells expressed IgM, IgA, and IgG, with IgG being dominant and known to be more potent at neutralizing SARS-CoV-2 than IgA [31]. The decline in IgA$^+$ cells over time may indicate either waning of the systemic response or redistribution to mucosal sites. Consistent with previous findings, IgA1 and IgG1 were the most prevalent subclasses [32]. Importantly, our data reveal class switching toward IgG4 six months after primary mRNA-based COVID-19 vaccination, extending our earlier findings in ESRD patients after repeated vaccination [9]. This delayed increase of IgG4 is consistent with the slow kinetics of IgG4 recombination, determined by the genomic order of heavy-chain genes (γ3–γ1–γ2–γ4) [33]. It has been reported after repeated mRNA vaccination in infection-naive healthy individuals [16,17], but not after adeno-viral vector vaccines, suggesting a unique feature of mRNA-based platforms [7,8,15,34]. The inverse correlation between IgG4 and IgG1 likely reflects inter-individual reciprocal patterns, where participants with higher IgG4 frequencies tended to have relatively lower IgG1, even though median IgG1 levels remained stable over time [35]. The decline of IgG2 and IgG3 likely reflects waning of early responses rather than precursors of IgG4, as no correlation with IgG4 was observed. The consistent delayed shift toward IgG4 across all groups, including controls, underscores that this represents a general feature of the mRNA-based COVID-19 vaccine response rather than a phenomenon restricted to patient subgroups.

Among the four of six KTRs with this delayed class switching toward S-binding IgG4, all received corticosteroids and a calcineurin inhibitor, and two additionally received MMF. Despite the suppressive effects of these agents—ranging from inhibition of cytokine synthesis and T-cell activation to reduced proliferation of lymphocytes [36,37]—IgG4 induction still occurred, indicating that class switching is possible under (combined) immunosuppression. This contrasts with recent findings suggesting that the use of MMF was associated with reduced IgG1 and IgG4 levels [14]. We previously observed increased IgG4 responses after vaccination in ESRD patients treated with other immunosuppressants such as azathi-oprine, sulfasalazine, or cyclophosphamide [9]. Notably, cytokines implicated in IgG4 induction include IL-4, IL-10, and IL-13, along with Th2 responses, although the precise mechanisms remain incompletely understood [8].

Whether the limited Fc-mediated functionality of IgG4 influences protective immune responses remains uncertain. We previously found that class switching toward IgG4 reduced Fc-mediated effector functions without impairing neutralization in ESRD patients [9]. While Fc functions contribute to antiviral protection [8,15,32], they may also exacerbate inflammation and disease severity [38,39]. However, recent data in healthy individuals showed that elevated IgG4 after three mRNA-based COVID-19 vaccine doses correlated with a higher risk of breakthrough infection, whereas IgG1 levels, C1q- and Fcγ receptor-binding capacity, and neutralization were associated with protection [40]. Of note, no SARS-CoV-2 infections were documented during the study.

This study provides insights into de novo IgG4 responses in KTRs, HD/PD and CKD patients. The modest subgroup sizes may have limited the detection of subtle differences, and effect size estimates are inherently less stable in small samples and should therefore be interpreted with caution. However, this approach allows assessment of not only sta-tistical significance but also potential biologically relevant shifts, even in limited samples. Furthermore, no correction for multiple testing was applied, which may have increased the risk of type I error. Next, the absence of functional assays precludes conclusions on whether the observed subclass distributions translate into altered effector functions in vivo. The findings should be regarded as preliminary and need validation in larger cohorts incorporating functional antibody read-outs. Adequate sample sizes are especially important given that, in our study, approximately half of KTRs who mounted an antibody response lacked detectable S-binding B cells. Only KTRs with antibody and S-binding responses were included, which may have introduced selection bias and limits the generalizability to the broader KTR population. As this

was an exploratory study, our aim was to characterize IgG subclass dynamics and S-specific B-cell phenotypes in KTRs capable of mounting a measurable vaccine-induced immune response. Lastly, IgG4 expression was identified indirectly by inference (i.e., by exclusion of other subclasses); however, this approach is supported by previously reported detectable serum IgG4 titers [9]. Given that IgG4 is the least abundant IgG subclass [7], our initial gating strategy—focusing on IgG1, IgG2, and IgG3—was considered appropriate, particularly given the limited availability of fluorochrome-conjugated antibodies that could be distinguished on our flow cytometer. In hindsight, including a direct IgG4 marker would have been preferable. This requires careful interpretation, as IgG4 B cell frequencies may have been underestimated due to the absence of a specific IgG4 antibody in the panel.

Our findings demonstrate that mRNA-based COVID-19 vaccines can induce durable virus-specific B-cell responses among antibody responders, with S-binding B cells detected only in a subset of immunocompromised KTRs, where they displayed characteristics similar to those in dialysis patients, CKD patients, and controls. We provide evidence for a delayed shift toward IgG4 after two vaccine doses across all groups, indicating that IgG4 class switching can occur despite severe kidney dysfunction or use of immunosuppressive therapy. The clinical implications of this shift remain uncertain, highlighting the need for larger studies to clarify its mechanistic basis, functional correlates, and potential impact on protection and disease severity.

## Supporting information

**S1 Fig. Flow cytometry gating strategy.**
(PDF)

**S2 Fig. Pie chart representation of B cell subsets — naïve (IgD$^+$CD27$^-$), non-switched memory (IgD$^+$CD27$^+$), switched memory (IgD$^-$CD27$^+$), and double negative (IgD$^-$CD27$^-$) — shown as fractions of total CD19＋B cells (100%) over time across all study groups.** Samples were analyzed at baseline (V1), 28 days post-vaccination (V3), and 6 months post-vaccination (V4).
(PDF)

**S3 Fig. Pie chart representation of S-binding plasmablasts (Ki67$^+$) and plasma cells (Ki67$^-$), shown as fractions of antibody-secreting cells (ASCs; 100%) over time in all study groups.** Samples were analyzed at 28 days post-vaccination (V3), and 6 months post-vaccination (V4).
(PDF)

**S4 Fig. Flow cytometry gating in KTRs with relatively high S-binding IgG4 percentages over time.** Samples were analyzed at 28 days post-vaccination (V3, left) and 6 months post-vaccination (V4, right). For each participant, the top row shows total B cells and the bottom row shows S-binding B cells. (A) KTR2; (B) KTR3; (C) KTR4; (D) KTR6.
(PDF)

**S1 Table. Monoclonal antibodies (mAbs) used in B cell panel.**
(PDF)

**S2 Table. Immunosuppressive agents received per patient in the KTR group, including excluded participants.** White bars represent included KTRs, light grey bars indicate excluded KTRs with negligible S-binding B-cell counts, and dark grey bars indicate individuals without antibody response who served as negative controls.
(PDF)

**S3 Table. Medians, IQRs, and full statistical comparisons of S1 IgG antibody levels and S-binding B cell frequencies over time.**
(PDF)

**S4 Table. Medians, IQRs, and full statistical comparisons of circulating ASCs and CXCR3$^+$ B cells within S-binding and switched memory B cell populations.**
(PDF)

**S5 Table. Medians, IQRs, and full statistical comparisons of frequencies of IgM$^+$, IgA$^+$, and IgG$^+$ B cells within S-binding and switched memory B cell populations.**
(PDF)

**S6 Table. Medians, IQRs, and full statistical comparisons of frequencies of IgA$^+$ and IgG$^+$ S-binding B cell subclasses and correlations of IgG4 with other IgG subclasses.**
(PDF)

**S7 Table. Medians, IQRs, and full statistical comparisons of heatmaps showing IgA and IgG subclasses (total 100%) within S-binding and switched memory B cell populations.**
(PDF)

## Acknowledgments

We would like to thank T. Standaar and I. Moerman (Amsterdam UMC location University of Amsterdam, Renal Transplant Unit, Meibergdreef 9, Amsterdam, the Netherlands) for their help with participant enrollment, and T. Dekker (Amsterdam UMC, University of Amsterdam, Department of Experimental Immunology, Amsterdam Infection & Immunity, Meibergdreef 9, Amsterdam, the Netherlands) for her help with running the Biobank Renal Diseases of the Amsterdam UMC location AMC. We would like to thank Gius Kerster and Wouter Olijhoek (Amsterdam UMC, location University of Amsterdam, Department of Medical Microbiology and Infection prevention, Amsterdam, the Netherlands; Amsterdam institute for Infection and Immunity, Infectious diseases, Amsterdam, the Netherlands) for their help in S-protein production and quality control. We would also like to thank all participants for contributing to this study.

## Author contributions

**Conceptualization:** Sophie C. Frölke, Marit van Gils, Mathieu Claireaux, Michiel C. van Aalderen, Ester B. M. Remmerswaal.

**Data curation:** Sophie C. Frölke, Kenney G. Amirkhan, Nelly van der Bom-Baylon, Marit van Gils, Mathieu Claireaux, Ester B. M. Remmerswaal.

**Formal analysis:** Sophie C. Frölke, Ester B. M. Remmerswaal.

**Funding acquisition:** A. Lianne Messchendorp.

**Investigation:** Sophie C. Frölke, Ester B. M. Remmerswaal.

**Methodology:** Sophie C. Frölke, Dimitri A. Diavatopoulos, Michiel C. van Aalderen, Ester B. M. Remmerswaal.

**Project administration:** A. Lianne Messchendorp, Ester B. M. Remmerswaal.

**Resources:** Ester B. M. Remmerswaal.

**Software:** Sophie C. Frölke, Ester B. M. Remmerswaal.

**Supervision:** Michiel C. van Aalderen, Ester B. M. Remmerswaal, Frederike J. Bemelman.

**Validation:** Sophie C. Frölke, Ester B. M. Remmerswaal.

**Visualization:** Sophie C. Frölke, Michiel C. van Aalderen, Ester B. M. Remmerswaal, Frederike J. Bemelman.

**Writing – original draft:** Sophie C. Frölke, Michiel C. van Aalderen, Frederike J. Bemelman.

**Writing – review & editing:** Sophie C. Frölke, Kenney G. Amirkhan, Marit van Gils, Mathieu Claireaux, Suzanne E. Geerlings, Rory D. de Vries, Jan-Stephan F. Sanders, Luuk B. Hilbrands, Dimitri A. Diavatopoulos, A. Lianne Messchendorp, Michiel C. van Aalderen, Frederike J. Bemelman.

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
