## [Decision Letter · Decision Letter 0]

22 Dec 2025

PONE-D-25-54840Class switching toward IgG4 six months after primary mRNA-based COVID-19 vaccination in kidney patientsPLOS One

Dear Dr. Frölke,

Thank you for submitting your manuscript to PLOS ONE. After careful consideration, we feel that it has merit but does not fully meet PLOS ONE’s publication criteria as it currently stands. Therefore, we invite you to submit a revised version of the manuscript that addresses the points raised during the review process.

The manuscript is well-written and provides interesting insights into the COVID vaccine's response in kidney transplant patients. Please address reviewers' comments. Also, please consider: - Explain the clinical significance of your findings - Did you find that the responders were less likely to get infected? What was the infection and re-infection rate after vaccination on your population? - Did the IgG4 antibodies show to be of better protection than other subtypes in your population?

We look forward to receiving your revised manuscript.

Kind regards,

Maria Lourdes Gonzalez Suarez, MD, PhD

Academic Editor

PLOS One

Additional Editor Comments:

The manuscript is well-written and provides interesting insights into the COVID vaccine's response in kidney transplant patients.

Please address reviewers' comments.

Also, please consider:

- Explain the clinical significance of your findings

- Did you find that the responders were less likely to get infected? What was the infection and re-infection rate after vaccination on your population?

- Did the IgG4 antibodies show to be of better protection than other subtypes in your population?

Thank you

Reviewers' comments:

Reviewer's Responses to Questions

**Comments to the Author**

1. Is the manuscript technically sound, and do the data support the conclusions?

Reviewer #1: Yes

Reviewer #2: Yes

2. Has the statistical analysis been performed appropriately and rigorously? 

Reviewer #1: Yes

Reviewer #2: Yes

3. Have the authors made all data underlying the findings in their manuscript fully available?

Reviewer #1: Yes

Reviewer #2: Yes

4. Is the manuscript presented in an intelligible fashion and written in standard English?

Reviewer #1: Yes

Reviewer #2: Yes

5. Review Comments to the Author

Reviewer #1: Thank you for the opportunity to review this manuscript. This work addresses an important topic, the dynamics of IgG subclass responses following mRNA COVID-19 vaccination in immunocompromised kidney patients. The study is timely, relevant, well written and methodologically sound and provides meaningful exploratory insights.

I have some minor recommendations to be addressed before the manuscript is suitable for publication.

1. The Sentence “Nearly half of KTRs…” appears to be repeated in line 336 and 340, please remove one.

2. As the classification of IgG4 was exclusionary, despite the justification provided (Frölke et al., manuscript submitted), consider addressing it by providing a brief quantitative support (correlation coefficients) and expand on “careful interpretation” by maybe describing the limitations of a potential misclassification for clearer framing.

3. I would recommend rechecking the effect sizes. I would also recommend adding a sentence emphasizing that the effect sizes and p-values should be interpreted with caution due to small size and multiple comparisons.

4. Some results are repeated almost word for word in the Discussion. Revising for conciseness would improve flow.

5. I would clarify (line 375) that IgG4 occurred, but it did not occur in all (4/6). This is important since this was already a selected subset for more immunocompetent KTRs.

6. Minor consistency edits: “The Netherlands vs the Netherlands” (line 25), “Intern Med vs Internal Medicine” (line 23), and consistent use of “+” either as text or superscript.

Reviewer #2: The study was performed in the participants of the RECOVAC Immune response study and examines evaluates spike specific humoral immunity and IgG subclass switching toward spike specific IgG4 antibodies following a two-dose mRNA 1273 COVID 19 vaccine administered 28 days apart among kidney transplant recipients, dialysis patients (HD and PD), advanced CKD and non-immunocompromised controls that were partners/siblings or household members of participant group. Antibody levels were measured prior to vaccine, at 1- and 6-months post second dose. Approximately half of kidney transplant recipients lacked detectable spike-binding B cells after two mRNA1273 doses, while responders across all groups showed persistent spike-specific B cells with a delayed relative shift from IgG1 toward IgG4 by six months.

While the premise of the study is interesting, there have been prior studies documenting the diminished response to vaccines in CKD and kidney transplant patients; the study has several limitations which the authors do acknowledge however they may not be addressed in its current form.

Some of the limitations are as follows:

1) Sample size: Highly selective sample size and relatively small proportion of patients selected, only responders among kidney transplant recipients were selected which limits generalizability among transplant recipients

2) Limited novelty: IgG4 class switching after mRNA vaccinations have been looked at among kidney transplant patients in a study by Juarez et al in a larger cohort previously.

3) Statistical concerns: Multiple subgroup testing with a limited sample size

4) Clinical utility and relevance of the findings are limited, diminished vaccine response among CKD/ESRD as well as transplant patients are well established.

6. PLOS authors have the option to publish the peer review history of their article (what does this mean? ). If published, this will include your full peer review and any attached files.

**Do you want your identity to be public for this peer review?** For information about this choice, including consent withdrawal, please see our Privacy Policy .

Reviewer #1: No

Reviewer #2: No

---

## [Author Response · Author response to Decision Letter 1]

5 Feb 2026

Point-by-point response to reviewers of PLOS One

Date: 5th of February, 2026

Editor comments

The manuscript is well-written and provides interesting insights into the COVID vaccine's response in kidney transplant patients.

Please address reviewers' comments.

Also, please consider:

- Explain the clinical significance of your findings

Answer: We thank the Editor for emphasizing the importance of explaining the clinical significance of our findings. Following primary mRNA-based COVID-19 vaccination, we observed a delayed shift toward IgG4 across all study groups, indicating that the mRNA platform can induce IgG4 class switching after only two vaccinations, even in the context of advanced kidney disease and immunosuppressive therapy.

As reported in the Discussion section, page 20, lines 389-393, using the same flow cytometric B-cell panel combined with functional antibody assays, we previously demonstrated that IgG4 class switching was associated with reduced Fc-mediated effector functions, while neutralizing capacity remained preserved, in patients with advanced kidney disease.1 While Fc functions contribute to antiviral protection, they may also exacerbate inflammation and disease severity. Our current findings extend these observations to kidney transplant recipients (KTRs), suggesting that IgG4 skewing may represent a shared immunological feature of mRNA vaccination (lines 375-378). However, the functional correlates and clinical implications of this in time evolving subclass distribution—particularly with respect to protection against infection and disease severity—require confirmation and further investigation in larger, adequately powered cohorts incorporating functional antibody readouts. Adequate sample sizes are especially important given that, in our study, approximately half of KTRs who mounted an antibody response lacked detectable S-binding B cells. Notably, as reported in lines 393-396, a recent study in healthy individuals have reported that elevated IgG4 levels following three mRNA-based COVID-19 vaccine doses were associated with an increased risk of breakthrough infection, whereas IgG1 levels, C1q binding, Fcγ receptor engagement, and neutralization capacity correlated with protection.2 Together, these data highlight the need to better understand how IgG subclass skewing may influence long-term vaccine efficacy and clinical outcomes, especially in immunocompromised patients.

The Discussion section has been updated to incorporate this information (page 21, lines 402–406), supplementing information already presented as mentioned above and in the Abstract, page 4, lines 69-70, and the Discussion section, page 21-22, lines 428-430.

- Did you find that the responders were less likely to get infected? What was the infection and re-infection rate after vaccination on your population?

Answer: All participants were seronegative for S1-specific IgG antibody responses at baseline and seropositive after vaccination, as per the inclusion criteria. During the study, SARS-CoV-2 infection was defined as either a self-reported positive PCR test or the presence of nucleocapsid-specific antibodies (this is added to the Methods section, page 6, lines 116-117). No SARS-CoV-2 infections were documented during the study (this is added to the Discussion section, page 20, line 396).

At V3 (28 days after the second vaccination), four KTRs (KTR2, KTR3, KTR7, and KTR10) underwent SARS-CoV-2 testing in March–April 2021, all with negative results. At V4 (6 months after the second vaccination), no participants reported undergoing SARS-CoV-2 testing.

Extended follow-up data were collected as part of a separate RECOVAC project through a questionnaire. Two participants reported a positive SARS-CoV-2 test: KTR6 (PCR test via public health services on 15-10-2021) and KTR10 (self-test on 01-12-2021). Both reported symptomatic infection, but neither required hospital admission.

Notably, all SARS-CoV-2 testing and both reported infections occurred in KTRs, who may have been particularly vigilant regarding testing due to awareness of their increased risk.3 Both participants had positive antibody responses, but whereas KTR6 had detectable S-binding B-cell responses, KTR10 had negligible S-binding B-cell counts. Given the small number of infections and the limited follow-up data, no conclusions can be drawn regarding differences in infection or reinfection risk between S-binding responders and non-responders.

When the Editor wishes to add this data to the manuscript or supporting information we are of course willing to do so.

- Did the IgG4 antibodies show to be of better protection than other subtypes in your population?

Thank you

Answer: With regard to this question, we kindly refer you to our response to your first comment, as functional antibody assays were not performed in this study. If “better protection” is interpreted as preserved neutralization capacity, our recent data in end-stage renal disease patients indicate that IgG4, similar to IgG1, contributes to neutralization, as described above.1

Comments to the Author

Review Comments to the Author

Reviewer #1: Thank you for the opportunity to review this manuscript. This work addresses an important topic, the dynamics of IgG subclass responses following mRNA COVID-19 vaccination in immunocompromised kidney patients. The study is timely, relevant, well written and methodologically sound and provides meaningful exploratory insights.

I have some minor recommendations to be addressed before the manuscript is suitable for publication.

1. The Sentence “Nearly half of KTRs…” appears to be repeated in line 336 and 340, please remove one.

Answer: We agree with the reviewer that one sentence needs to be removed. The adjustment has been made in the Discussion section on page 19 in line 348.

2. As the classification of IgG4 was exclusionary, despite the justification provided (Frölke et al., manuscript submitted), consider addressing it by providing a brief quantitative support (correlation coefficients) and expand on “careful interpretation” by maybe describing the limitations of a potential misclassification for clearer framing.

Answer: We detected a fraction of IgG B cells that was negative for IgG1, IgG2, and IgG3, which we interpreted as IgG4. The classification of IgG4 was therefore exclusionary, and we agree that this warrants further discussion. Given that IgG4 is the least abundant IgG subclass,4 our initial gating strategy—focusing on IgG1, IgG2, and IgG3—was considered appropriate, particularly given the limited availability of fluorochrome-conjugated antibodies that could be distinguished on our flow cytometer. In hindsight, including a direct IgG4 marker would have been preferable. However, this fraction within the S-binding IgG B cell population has previously been shown to correspond (no correlation analysis was performed) with detectable serum IgG4 titers, thereby justifying its use for the detection of S-binding IgG4 B cells (stated in the Methods section, page 9, lines 144-149).1 This provides quantitative support that the exclusionary gating method reliably detects the IgG4 fraction.

We agree with the reviewer that careful interpretation is warranted. Potential limitations include the possibility that a subset of IgG4 B cells may have been missed due to the lack of a specific IgG4 antibody in the panel. Consequently, the detected frequencies of IgG4 B cells may be underestimated and may be higher or more prevalent than observed. Therefore, our findings regarding IgG4 should be interpreted in the context of this methodological limitation.

This reasoning had been added to the Discussion section on page 21 in the following line(s): 411-418.

3. I would recommend rechecking the effect sizes. I would also recommend adding a sentence emphasizing that the effect sizes and p-values should be interpreted with caution due to small size and multiple comparisons.

Answer: As suggested by the reviewer, effect sizes have been rechecked multiple times. We would like to refer the reviewer to the following text in the Discussion section (page 20-21, lines 397–402), where we emphasize that both effect sizes and p-values should be interpreted with caution due to the small sample size and multiple comparisons.

4. Some results are repeated almost word for word in the Discussion. Revising for conciseness would improve flow.

Answer: We agree with the reviewer that the Discussion must be revised for conciseness to improve flow. The following within the Discussion section was adjusted:

- page 19, line 348;

- page 19, line 362.

5. I would clarify (line 375) that IgG4 occurred, but it did not occur in all (4/6). This is important since this was already a selected subset for more immunocompetent KTRs.

Answer: The adjustment has been made in the Discussion section on page 20 in the following line: 379.

6. Minor consistency edits: “The Netherlands vs the Netherlands” (line 25), “Intern Med vs Internal Medicine” (line 23), and consistent use of “+” either as text or superscript.

Answer: The following adjustments have been made:

- page 1-2 in the following lines: 16, 17, 19, 22, 23, 24, 34;

- page 3 line 62;

- page 11 line 184;

- page 23 lines 433, 435, 438, 439.

Reviewer #2: The study was performed in the participants of the RECOVAC Immune response study and examines evaluates spike specific humoral immunity and IgG subclass switching toward spike specific IgG4 antibodies following a two-dose mRNA 1273 COVID 19 vaccine administered 28 days apart among kidney transplant recipients, dialysis patients (HD and PD), advanced CKD and non-immunocompromised controls that were partners/siblings or household members of participant group. Antibody levels were measured prior to vaccine, at 1- and 6-months post second dose. Approximately half of kidney transplant recipients lacked detectable spike-binding B cells after two mRNA1273 doses, while responders across all groups showed persistent spike-specific B cells with a delayed relative shift from IgG1 toward IgG4 by six months.

While the premise of the study is interesting, there have been prior studies documenting the diminished response to vaccines in CKD and kidney transplant patients; the study has several limitations which the authors do acknowledge however they may not be addressed in its current form.

Some of the limitations are as follows:

1) Sample size: Highly selective sample size and relatively small proportion of patients selected, only responders among kidney transplant recipients were selected which limits generalizability among transplant recipients

Answer: The reviewer is correct that only KTRs who mounted both an antibody response and detectable S-binding B-cell responses were included, which may introduce selection bias and limits the generalizability of our findings to the broader KTR population. This limitation is explicitly acknowledged in the Discussion section (page 21, lines 407–409).

As this was an exploratory study, no statistical methods were used to predetermine sample size, and our aim was to characterize IgG subclass dynamics and S-specific B-cell phenotypes in KTRs capable of mounting a measurable vaccine-induced immune response. Importantly, the observation that a substantial proportion of KTRs lacked detectable S-binding B cells, despite seroconversion, highlights the decreased immune response in this population and underscores the need for further studies in larger, adequately powered cohorts. In contrast, antibody and S-binding B-cell responses were consistently detected in CKD patients, dialysis patients, and non-immunocompromised controls, indicating that the KTR group reflects their unique immunological impairment rather than a general limitation of the study design.

This reasoning has been incorporated in the Discussion section on page 21 in the following line(s): 405-411.

2) Limited novelty: IgG4 class switching after mRNA vaccinations have been looked at among kidney transplant patients in a study by Juarez et al in a larger cohort previously.

Answer: The reviewer is correct that IgG4 class switching following mRNA vaccination in KTRs has previously been reported by Juárez et al. in a larger cohort, as cited in the Introduction section (page 5, lines 87–88). However, there are several important differences between that study and our findings.

In the study by Juárez et al., IgG4 induction in kidney transplant recipients was observed two months after a fourth mRNA-based COVID-19 vaccination, with immune responses primarily assessed 21 days after the second dose. In contrast, we demonstrate delayed IgG4 class switching occurring as late as six months after the primary two-dose mRNA-1273 vaccination regimen. Moreover, this delayed subclass shift was consistently observed not only in KTRs and dialysis patients, as reported by Juárez et al., but also in patients with advanced CKD and in controls without kidney disease. Additional information on this study has been added to the Introduction section (page 5, lines 88–89).

In addition, within our cohort, four of six KTRs who developed IgG4 responses did so despite ongoing immunosuppressive therapy with corticosteroids and calcineurin inhibitors, with two patients additionally receiving mycophenolate mofetil (MMF). This contrasts with the findings of Juárez et al., who reported an association between MMF treatment and reduced IgG1 and IgG4 levels. This consideration has been added to the Discussion section (page 20, lines 383–384).

Together, these differences highlight that our study provides complementary and novel insights into the timing and persistence of IgG4 class switching following mRNA vaccination.

3) Statistical concerns: Multiple subgroup testing with a limited sample size

Answer: We would like to refer the reviewer to the following text in the Discussion section (page 20-21, lines 397–402), where we emphasize that both effect sizes and p-values should be interpreted with caution due to the small sample size and multiple comparisons.

4) Clinical utility and relevance of the findings are limited, diminished vaccine response among CKD/ESRD as well as transplant patients are well established.

Answer:

We refer the reviewer to our response to the Editor’s first question, where the clinical significance of our findings is discussed in detail. While diminished vaccine responses in CKD/ESRD and transplant populations are well established, our study provides additional mechanistic insight by demonstrating a delayed shift toward IgG4 following the primary two-dose mRNA-1273 vaccination regimen. As discussed, the clinical implications of this IgG subclass skewing remain uncertain and warrant further investigation. Specifically, larger, well-powered studies are needed to clarify the underlying mechanisms, functional correlates, and potential impact of IgG4 predominance on protection against infection and disease severity. Such studies may ultimately inform vaccination strategies in immunocompromised patients, including the optimal timing and number of booster doses.

Notably, concerns have been raised regarding IgG4 class switching and the potential development of IgG4-related autoimmunity; no data are currently available indicating such autoimmunity after two vaccine doses. Available evidence suggests that the observed IgG4 shift is benign and reflects normal immune adaptation to repeated antigen exposure.4

Accordingly, continued monitoring of vaccine-induced antibody profiles remains important.

If deemed appropriate by the reviewer and the Editor, we would be willing to incorporate this consideration into the manuscript or the Supporting Information.

References

1 Frolke, S. C. et al. SARS-CoV-2-specific B cell responses in non-draining lymph nodes and antibody functionalities in immunized end-stage renal disease patients. Sci Rep 15, 44266 (2025). https://doi.org/10.1038/s41598-025-27815-y

2 Martin Perez, C. et al. Post-vaccination IgG4 and IgG2 class switch associates with increased risk of SARS-CoV-2 infections. J Infect 90, 106473 (2025). https://doi.org/10.1016/j.jinf.2025.106473

3 Frolke, S. C. et al. Adherence to preventive measures after SARS-CoV-2 vaccinatio

---

## [Editor Report · Decision Letter 1]

9 Feb 2026

Class switching toward IgG4 six months after primary mRNA-based COVID-19 vaccination in kidney patients

PONE-D-25-54840R1

Dear Dr. Frölke,

We’re pleased to inform you that your manuscript has been judged scientifically suitable for publication and will be formally accepted for publication once it meets all outstanding technical requirements.

Kind regards,

Maria Lourdes Gonzalez Suarez, MD, PhD

Academic Editor

PLOS One

Additional Editor Comments (optional):

Thank you for addressing our comments.
---

## [Editor Report · Acceptance letter]

PONE-D-25-54840R1

PLOS One

Dear Dr. Frölke,

I'm pleased to inform you that your manuscript has been deemed suitable for publication in PLOS One. Congratulations! Your manuscript is now being handed over to our production team.

Kind regards,

on behalf of

Dr. Maria Lourdes Gonzalez Suarez

Academic Editor

PLOS One